# Structural Characteristics and Sound Absorption Properties of Waste Hemp Fiber

**Duoduo Zhang, Xinghai Zhou, Yuan Gao and Lihua Lyu ***

School of Textile and Material Engineering, Dalian Polytechnic University, Dalian 116034, China
* Correspondence: lvlh@dlpu.edu.cn

**Abstract:** In order to realize high-efficiency and high-value recycling of waste hemp fibers, the macromolecular structure, the supramolecular structure, and the morphological structure of waste hemp fibers were investigated by using Fourier transform infrared spectroscopy, X-ray diffractometry, upright metallurgical microscopy, and scanning electron microscopy. According to its structural characteristics, the sound-absorbing mechanism of waste hemp fiber was analyzed, and the reason for the good sound-absorbing performance of waste hemp fiber was clarified. The acoustic impedance transfer function test was used to analyze and compare the sound-absorbing performance of waste hemp fiber and several other fiber aggregates that could be used in the field of sound-absorbing, and the sound-absorbing performance of a waste hemp fiber composite material was tested. The research revealed that: the sequence of sound-absorbing performance of several fiber aggregates was cotton fiber, waste hemp fiber, wool fiber, and polyester fiber; that waste hemp fiber had excellent high-frequency sound-absorbing performance, with a maximum sound absorption coefficient of 0.95; and that the maximum sound absorption coefficient of the waste hemp fiber composite reached 0.93. Therefore, the waste hemp fiber has excellent sound-absorbing properties and has high application value in the field of sound absorption.

**Keywords:** waste hemp fiber; microstructure; sound absorption performance; sound absorption mechanism

## 1. Introduction

From an epidemiological perspective, noise pollution is an invisible killer that endangers human health even though it is a physical pollutant that is confined to a specific region and has no aftereffects. Prolonged exposure to excessive noise will not only damage one's hearing, vision, digestive system, nervous system, endocrine system, and cardiovascular system but will also have a negative impact on one's daily life, such as increased stress, reduced work efficiency, disrupted sleep, and disturbed communication [1]. In this case, the suppression of noise is an urgent problem, and the use of sound-absorbing materials in the acoustic propagation path can effectively reduce noise.

Today, various sound-absorbing materials such as mineral wool, synthetic fibers, and polymer foams are widely used in the building and construction industry. In general, such synthetic and mineral materials are superior to natural fibers in terms of sound absorption, insulation, and fire protection [2]. However, these traditional sound-absorbing materials, such as fiberglass and mineral wool, may pose some potential health hazards to humans, such as inhalation of fibers and particles that can cause health problems by causing irritation and deposition in the alveoli of the lungs. In addition, they can also cause some pollution to the environment, as they tend to break and produce solid waste that is not easily degradable [3]. As people become more health-conscious and environmentally friendly, researchers are working to develop low-cost, non-polluting, healthy, and environmentally friendly sound-absorbing materials. Natural fibers are beginning to be used in sound absorption because of their low cost, degradability, easy availability, and environmental and good sound absorption properties.

Sakthivel et al. [4] developed an environmentally friendly acoustic nonwoven material using recycled cotton fiber and polyester fibers, and the results showed that the PET/waste cotton nonwoven mat absorbed more than 70% of the acoustic impedance with good acoustic performance. Sakthivel et al. [5] studied the use of recycled cotton and polyester fibers for the production of acoustic nonwoven composites. The values obtained for the absorption and noise reduction coefficients indicate that the recycled polyester fiber nonwoven composites have excellent sound absorption properties over the entire frequency range. Lyu et al. [6] prepared a nonwoven wallcovering from waste wool fibers and low melting point polyamide fibers by a combination of web formation and hot pressing methods. The sound absorption coefficient was greater than 0.91 and the noise reduction coefficient was 0.56 for this nonwoven wallcovering with good sound absorption performance under optimized process conditions. Murean et al. [7] obtained acoustic-absorbing materials by simple hot pressing of wool fibers. The material has very good sound absorption properties, with a coefficient of absorption value exceeding 0.7 in the frequency range of 800–3150 Hz. The results demonstrate that the sound absorption properties of wool fibers are comparable to mineral wool or recycled polyurethane foam. Yang et al. [8] studied the sound absorption properties of cotton and wool fiber nonwovens as well as pure polyester nonwoven, and the study showed that cotton fiber nonwovens are a good alternative to traditional polyester sound-absorbing materials. However, the sound absorption performance of wool fiber-based nonwovens is relatively poor. Therefore, it could be found that cotton and wool, two natural fibers, are commonly used in the preparation of sound-absorbing materials, while polyester fibers are traditional synthetic sound-absorbing materials.

Among all plant fibers, hemp is one of the most interesting because of its excellent properties, low cost, and high cellulose content [9]. The tetrahydrocannabinol (THC) in hemp has the characteristics of a hallucinogenic addiction, so it is listed as a drug in many countries and is forbidden to be planted. The THC level of various cannabis strains varies significantly, but according to the 2018 U.S. Farm Bill, the strains with a THC content of less than 0.3 percent are classified as industrial hemp and can be grown and used [10]. The advancement of science and technology has led to research on "industrial hemp", which has also gradually deepened. Because of its good moisture absorption and breathable properties, antibacterial properties, high tensile strength properties, radiation resistance, and biodegradable properties, it is known as the most promising green product in the 21st century. At least 47 nations throughout the world presently cultivate cannabis for business or research purposes, with the largest producers of industrial hemp currently being China, Canada, and France [11]. With the U.S. Farm Bill allowing for the legal production of cannabis and the huge market for cannabis products, the cannabis industry is growing very rapidly [12]. Global cannabis trade across all markets is expected to reach USD 8.1 billion by 2021, representing an 83% three-year growth rate during the forecast period. In addition, the worldwide market is anticipated to register a value of USD 89 billion by 2025, growing at a 52% compound annual rate due to the continued growth in textile products, foodstuffs, manufacturing uses, and cannabidiol CBD. Given the promising future of hemp, Europe has rapidly developed a strong market for hemp and CBD. With roughly 80% of the USD 1.7 billion global hemp textile market in 2019, China controlled the textile industry [13]. Hemp is a versatile crop used in a variety of products, including building materials, textile products, paper, foodstuffs, furniture, luxury markets, cosmetic products, and hygiene items, while the textile sector generally processes industrial hemp into fibers, fabrics, ropes, yarns, and home textiles [14–16]. The percentage of short fibers in hemp fiber is about 7.72%, and these short fibers are eliminated in the form of waste in the manufacture of hemp yarn and fabric, resulting in a waste of resources [17,18]. Given that hemp is a natural fiber, there are various uses for its textile waste, such as reinforcing materials and other matrix materials made of composite materials, to improve the utilization of hemp fiber and reduce resource waste, so as to achieve the development goals of peak carbon emission and carbon neutrality.

The structure of hemp fiber is quite different from most other fibers. There are various shapes of cavities in the cross-section, spiral patterns, connected holes, and cracks in the longitudinal surface [19], which make domestic and foreign scholars interested in the application of hemp fiber in the field of acoustics-related research. Table 1 lists the research on the use of hemp fibers in the field of sound absorption.

**Table 1.** A summary of studies of hemp fiber sound absorbers.

| Researcher | Raw Material | Key Findings |
| --- | --- | --- |
| Xu Fan et al. [20] | hemp, wool, cotton fibers, ramie, and flax | The sound absorption coefficients of hemp, wool, cotton fiber, ramie, and flax were compared in the middle and low frequency bands. |
| Jing Zhang [21] | hemp staple fibers and polypropylene fibers | Composite sound absorption properties of hemp/polypropylene fiber composites varied under different preparation conditions. |
| Santoni et al. [22] | hemp fiber | The effects of each treatment on the physical properties of the hemp fiber composites and their sound absorption coefficients were investigated. |
| Liao et al. [23] | hemp fiber | Summarized the methods for improving the sound absorption of hemp fiber materials. |
| Boominathan et al. [24] | *Sansevieria stuckyi*, banana, and hemp fiber | The nonwoven mat made with blended fibers had a higher maximum sound absorption coefficient than single fiber nonwoven mats. The higher the proportion of hemp fiber, the greater the thickness of nonwovens, and the greater the sound absorption coefficient. |

As shown in Table 1, hemp fibers have acoustic properties and can be used for the preparation of acoustic composites. These articles, however, only discuss the impact of the manufacturing process, material composition, bulk density, morphology, and structural parameters on the acoustic properties of hemp fibers and do not investigate the relationship between the structural characteristics of hemp fibers and their acoustic properties, failing to explain the underlying reasons for hemp fibers' good acoustic properties.

In summary, due to the high rate of hemp fiber staple, poor neatness, and weaving difficulties, its application and development in textiles and apparel are somewhat constrained. The existence of hemp fibers inside the shaped cavity, the surface of the holes and gaps, and fiber curl twisted together will form a large number of interconnected pores. These characteristics give hemp fiber a unique sound-absorbing function. At the same time, hemp fiber, as a typical porous material, has sound-absorbing material that contains a variety of internal channels and cavities; sound waves can penetrate the structure, and the viscous effect and heat transfer caused by the loss of energy in the material absorption is the main mechanism of fiber material sound absorption [25]. However, the current research on the application of hemp fibers in the field of sound absorption is minimal; there is no detailed exploration of the internal structure of hemp fibers, and there is no clear work on the internal structure of hemp fibers and its sound absorption properties.

In order to provide a new research direction for the recycling of hemp fiber and provide the theoretical and experimental basis for the preparation of hemp fiber sound-absorbing composites, this paper took waste hemp fiber (long hemp spinning process generated by the fallen hemp fiber) as the research object, analyzed the relationship between the macromolecular structure of waste hemp fiber, the supramolecular structure, and their morphological structure, and their sound-absorbing properties. The sound-absorbing properties of fiber aggregates and their causes were studied. It was determined that the waste hemp fibers could be used for the preparation of sound-absorbing composites.

## 2. Experimental Details

### 2.1. Materials

Waste hemp fiber (long hemp spinning process generated by the fallen hemp fiber, Tongxiang Shangde Textile Co., Ltd., Jiaxing, China), cotton fiber (Mingxin Cotton Industry Co., Ltd., Shangqiu, China), wool fiber (Linyi Luozhuang Fudi Wool Textile Factory, Shandong, China), Polyester (Qinhuangdao Jinfang Chemical Fiber Co., Ltd., Qinhuangdao, China), and Polylactic acid (PLA) fiber (Quanzhou Smatin Import & Trade Co., Ltd., Quanzhou, China, melting point is 130 °C) were employed.

### 2.2. Equipment

An MP2000D Shanghai precision analytical balance (Changzhou No.1 Textile Equipment Co., Ltd., Changzhou, China), E-Max high-energy ball mill (Fuld Instruments and Equipment Co., Ltd., Shanghai, China), Spectrum One-B Fourier transform infrared spectrometer (Perkin Elmer Co., Ltd., Norwalk, CT, USA), D/max-3B X-ray diffractometer (Shimadzu, Kyoto, Japan), Upright metallurgical microscope (Changzhou No.1 Textile Equipment Co., Ltd., Changzhou, China), JSM-6460LV scanning electron microscope (SEM; JEOL Ltd., Beijing, China), DSCa-01 Digital sample carding machine (Jiacheng Mechanical & Electrical Equipment Co., Ltd., Tianjin, China), QLB-50D/Q Type Flat Vulcanizing Press Machine (Zhongkai Rubber & Plastic Machinery Co., Ltd., Wuxi, China), and SW422/SW477 impedance tube sound absorption test system (Prestige Acoustics Technology Co., Ltd., Beijing, China) were employed.

### 2.3. Sample Preparation

#### 2.3.1. Preparation of Fiber Aggregates

In order to compare the sound-absorbing performance of cotton fiber, wool fiber, waste hemp fiber, and polyester fiber, loose fibers of the same mass and different types were added to the impedance tube separately, and the ports of the impedance tube were sealed with gauze, thus ensuring that the thickness, volume, and density of all fiber aggregate samples were the same.

#### 2.3.2. Preparation of Waste Hemp Fiber Composite Material

The waste hemp fiber/PLA fiber composites were prepared by the hot pressing method using waste hemp fiber and polylactic acid (PLA) fiber as raw materials. The hot pressing temperature of the composite was set at 140 °C because PLA fibers had a melting point of 130 °C and degradation occurs at high temperatures, resulting in reduced performance. When the hot pressing time was short, the composite material was not fully molten and mixed. If the hot pressing time was too long, PLA would degrade, so the hot pressing time was set to 15 min. The hot pressing pressure could promote the flow of PLA fibers and facilitate the coating of waste hemp fibers with the PLA matrix, but when the hot pressing pressure was too high, the middle cavity of the fibers would be squeezed and become smaller, and the porosity of the composite material would be reduced, thus leading to a decrease in the sound absorption performance of the composite material. Therefore, the hot pressing pressure was set at 10 MPa. The waste hemp fiber and PLA fiber were mixed and carded in the carding machine at a ratio of 5:5 by mass to obtain a uniformly mixed preform, and then a certain quantity of the preform was added into the molds with diameters of 30 mm and 100 mm and heights of 10 mm, fed into the preheated plate vulcanization press, and hot-pressed at a temperature of 140 °C, and a pressure of 10 MPa for 15 min. The waste hemp/PLA fiber composite material was obtained after cooling, shaping, and demolding, as shown in Figure 1.

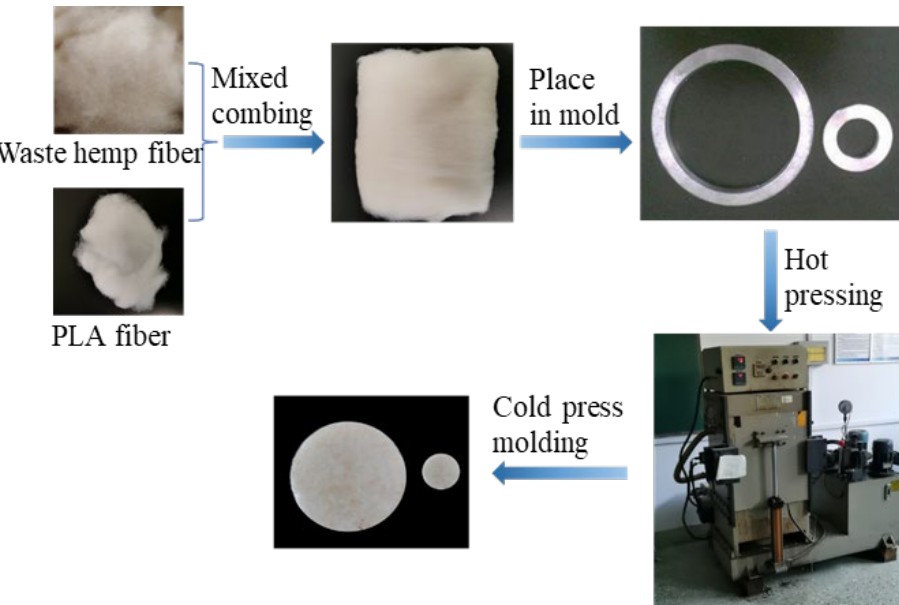

**Figure 1.** Preparation of waste hemp/PLA fiber composite material.

*2.4. The Structural Characteristics and Test Method of Waste Hemp Fiber*

2.4.1. Macromolecular Structure

A specific quantity of waste hemp fiber was ground in an E-Max high-energy ball crusher, and the resulting fiber powder was mixed with the treated potassium bromide before being pressed into tablets and molded. The molecular structure was examined using a Fourier transform infrared spectrometer.

2.4.2. Supramolecular Structure

The E-Max high-energy ball mill was used to grind a fixed amount of waste hemp fiber into powder, which was then examined using X-ray diffraction. The peak strength method [26] was used to calculate the crystallinity of waste hemp fiber, and the formula is provided below.

$$X_C = \frac{I_{002} - I_{am}}{I_{002}} \tag{1}$$

where: $X_C$ is crystallinity, $I_{002}$ is the maximum intensity of the lattice diffractive peak, $I_{am}$ is the diffractive intensity of the amorphous phase, and the background diffractive intensity is $20°$.

2.4.3. Morphological Structure

Length testing of waste hemp fibers—Hand arrangement was used to arrange discarded waste hemp fibers into fiber bundles with a parallel longitudinal orientation, straight ends, and one end that was flush. A total of 100 waste hemp fibers were gauged to determine the fiber length range.

Fineness testing of waste hemp fibers—The diameter of waste hemp fiber was tested with an upright metallographic microscope according to NY-T 2338-2013 "Micro-image Method for Rapid Detection of Flax Fiber Fineness". The number of waste hemp fibers tested was 100, and the results were averaged.

Observation of the transverse and longitudinal morphology of waste hemp fibers—A scanning electron microscope was used to examine the transverse and longitudinal appearance of waste hemp fiber. Before the test, the waste hemp fiber was fixed, and gold was coated with a sputtering coater to provide conductivity and eliminate charging artifacts.

*2.5. Parameter Calculation and Sound Absorption Test of Fiber Aggregates*

2.5.1. Calculation of Volume Fraction and Bulk Density of Fiber Aggregates

Density of fiber aggregates—The density of fiber aggregates was determined using a gradient column. The method was based on comparing the immersion depth of the fiber spheres being tested relative to the density standard of glass spheres in a cylinder with liquid, and the density of the fiber could be found by checking the density height graph according to the height of the fiber spheres.

Volume fraction of fiber aggregates—Four kinds of fibers—cotton fiber, wool fiber, waste hemp fiber, and polyester fiber—were selected and prepared into fiber aggregates of 10 cm in diameter, 1 cm in thickness, and 78.54 cm$^3$ in volume, all of which had the same bulk density. Because the densities of the four kinds of fibers were different, the volume fraction of the fibers was calculated as shown in Equation (2).

$$V_f = \frac{m/\rho}{V_0} \times 100\% \qquad (2)$$

where: $V_f$ is the fiber volume fraction (%); $m$ is the mass of the fiber aggregate (g); $\rho$ is the fiber density (g/cm$^3$); $V_0$ is the volume of the fiber aggregate (cm$^3$).

2.5.2. Sound Absorption Performance Test

According to the standard GB/T 18696.1-2004 "Measurement of acoustic impedance tube absorption coefficient and acoustic impedance", the absorption coefficients of samples at different frequencies were tested under the conditions of an atmospheric temperature of 22 °C and relative humidity of 68% using an SW422/SW477 impedance tube sound absorption test system. The measured sound absorption coefficient was the average of six measurements, and then the sound absorption coefficient curve of the materials under different frequencies was drawn. The sound absorption coefficient test is shown in Figure 2.

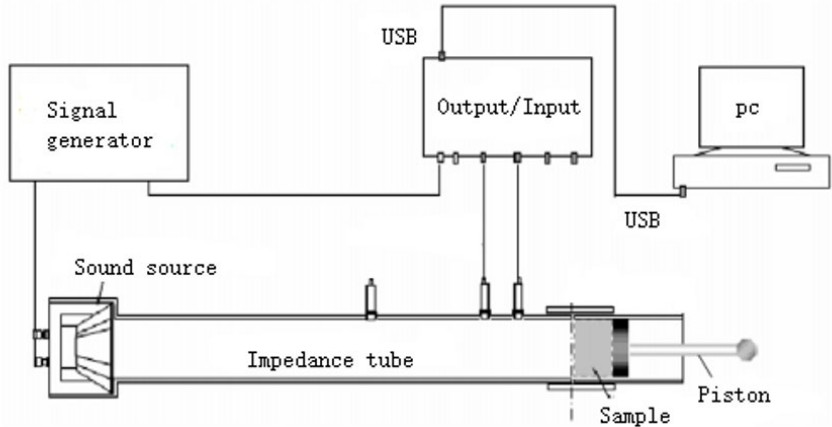

**Figure 2.** Schematic diagram of the sound absorption test.

The ability of sound-absorbing materials is mainly determined by the average sound absorption coefficient ($\alpha$) and noise reduction coefficient (*NRC*). For materials, when the average sound absorption coefficient is greater than 0.2, it can be called a sound-absorbing material. When the average sound absorption coefficient is greater than 0.56, they are known as high-efficiency sound-absorbing materials [27]. The average sound absorption coefficient refers to the average value of the sound absorption coefficient measured at the six frequencies of 125 Hz, 250 Hz, 500 Hz, 1000 Hz, 2000 Hz, and 4000 Hz. The average absorption coefficient calculation formula is below:

$$\alpha = \frac{\alpha_{125} + \alpha_{250} + \alpha_{500} + \alpha_{1000} + \alpha_{2000} + \alpha_{4000}}{6} \qquad (3)$$

Additionally, when the *NRC* is more significant than or equal to 0.2, the material can also be called a sound-absorbing material [28]. The *NRC* was the average sound absorption coefficient at 250 Hz, 500 Hz, 1000 Hz, and 2000 Hz frequencies. The calculation method of *NRC* is shown below:

$$NRC = \frac{\alpha_{250} + \alpha_{500} + \alpha_{1000} + \alpha_{2000}}{4} \tag{4}$$

## 3. Results and Discussion

### 3.1. Structural Characteristics of Waste Hemp Fiber

### 3.1.1. Macromolecular Structure

To investigate the relationship amongst chemical structure and sound-absorbing capabilities of waste hemp fiber, a Fourier transform infrared spectrometer was used to evaluate the spectra of waste hemp fiber, as seen in Figure 3, and the attribution and explanation of infrared peaks are shown in Table 2.

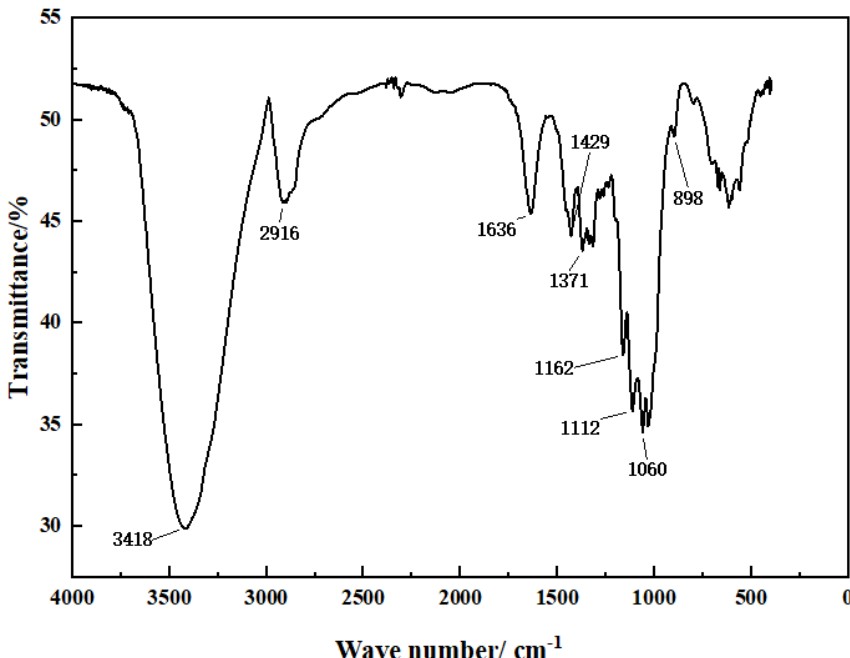

**Figure 3.** Infrared spectroscopy of waste hemp fiber.

**Table 2.** Infrared spectral analysis results of waste hemp fiber.

| Wavenumber/cm$^{-1}$ | Characteristic Peak Attribution | Characteristic Peak |
|---|---|---|
| 3418 | O–H telescopic vibration | Lignin and cellulose |
| 2916 | C–O telescopic vibration | Lignin |
| 1636 | C=O telescopic vibration | Lignin |
| 1429 | CH$_2$ bending vibration | Cellulose and lignin |
| 1371 | CH bending vibration | Cellulose and hemicellulose |
| 1162 | C–O–C telescopic vibration | Cellulose and lignin |
| 1060 | C=O telescopic vibration | Cellulose and hemicellulose |
| 898 | β-glucosidic bond vibration | Carbohydrate characteristic peak |

From the infrared spectrum, it has been shown that waste hemp fiber was largely constituted of cellulose, hemicellulose, and lignocellulose, and cellulose was indeed the main component of waste hemp fiber [29]. Cellulose is the most abundant natural polymer carbohydrate on the Earth, and it is a linear polysaccharide composed of D-glucose units (dehydrated glucose units) connected by β-1,4-glucoside. Adjacent glucose rings in the crystallization region are mutually inverted, the hydrogen atoms in the glucose are in the

vertical direction of the oxygen six-ring chair structure plane, and the hydroxyl groups are distributed on both sides of the chair structure plane. The molecular formula is $(C_6H_{10}O_5)_n$, as is the structural formula [30,31]. When sound waves were generated and incident on the surface of the waste hemp fiber, it caused the movement of carbon atoms and oxygen atoms in glucose inside the fiber. However, due to the different masses, there was a difference in movement, which caused the friction effect to form a friction force, hindered the movement of the atoms, and caused the sound energy to be converted into heat energy to dissipate.

The main chain of cellulose macromolecules had a large number of hydroxyl groups, allowing hydrogen bonds and van der Waals interactions to form among both adjoining cellulose macromolecules via the hydroxyl groups on the glucose and the oxygen atoms on adjacent glucose rings or the oxygen atoms on glycosidic bonds, making the structure of the straight-chain macromolecules more stable. In the oxygen hexa-ring chair structure, there were two parallel triangular planes and a parallelogram plane with a particular angle at the same time, resulting in greater gaps and voids between macromolecular chain segments and inside the oxygen hexa-ring structure [32]. Whenever the sound wave was incident within the fiber, there were more gaps and voids between the adjacent cellulose macromolecule chain segments, and there were also interconnected voids inside the oxygen hexacyclic ring. The sound energy's propagation path became more sinuous and circuitous as a result of these gaps and spaces, and the sound energy was fully consumed and dispersed. Secondly, due to the strong bonding force of hydrogen bonds, the macromolecule chain segments could not move freely, and when the sound energy acted upon those macromolecule molecular chains, the major chain of the cellulose chains generated vibration, which drove the rotation of both the hydrogen bonds and other single bonds inside the molecular chains, thus creating more friction force between the molecular chain segments and finally dissipating the sound energy into the thermal and mechanical energy [33].

### 3.1.2. Supramolecular Structure

The X-ray diffraction pattern of the waste hemp fibers can be seen in Figure 4. From Figure 4, it has been shown that the diffraction peaks near 16.72° and 22.1° belonged to the (101) and (002) crystallographic planes of cellulose, respectively, which indicated that the waste hemp fiber was of the typical cellulose I crystal type [34]. The crystallinity of waste hemp fiber was calculated according to Equation (1) as 48.84%, whereas the crystallinity of cotton, wool fiber, and flax fiber was 78.8% [35], 81.2% [36], and 69.3% [37], respectively. As a result, discarded waste hemp fibers had a lower crystallinity than other natural fibers.

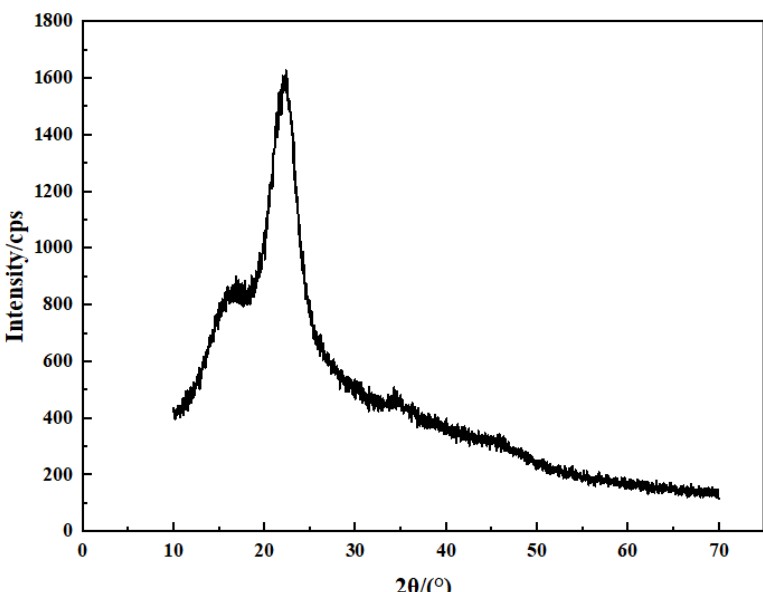

**Figure 4.** X-ray diffraction pattern of waste hemp fiber.

The crystal region with a regular and compact setup and a small proportion of gaps and holes in the molecular arrangement structure of waste hemp fiber was smaller than that of a disordered molecular chain arrangement, and there was a loose structure and a large proportion of voids and holes due to the poor crystallinity of waste hemp fiber. Sound waves were propagated in the molecular structure of fibers via the axial direction of the molecular backbone, the vibration of atoms in the molecular chain, and bond deformation [38]. Whenever the crystallization of the fiber lessened, the spacing between both molecules was higher, and the molecular chains were arranged loosely, resulting in increased porosity. As a result, the connection among both molecules was flimsy, chemical chains were easier to shift, and the relative motion was more likely to occur, resulting in increased sound wave consumption. Secondly, because of the differences in the degree of sparsity between the crystalline and non-crystalline areas within the waste hemp fiber, there were differences in temperature everywhere, which made the formation of temperature gradients between neighboring areas promote the transfer of heat, which in turn accelerated the rate of conversion of acoustic energy into heat.

### 3.1.3. Morphological Structure

Fiber length—The length of 100 waste hemp fibers was measured in the range of 20–75 mm, and the specific distribution is seen in Table 3; therefore the main length of waste hemp fibers is 20–40 mm. Hemp spinning could be divided into long hemp spinning and short hemp spinning, and hemp fiber with an average length of about 80 mm made by the combing process was used for long hemp spinning. Long hemp spinning mainly used the ramie and flax spinning processes, both of which are used in long fiber spinning; in the hemp long fiber spinning yarn carding process, a large number of short fibers below 40 mm fall and therefore produce a large number of fallen hemp fibers. According to the current technology level, every ton of long hemp yarn produced would produce about 1.3 tons of fallen hemp [39]. The short length and poor neatness of the fallen hemp fibers make them less spinnable and more difficult to spin into high-count yarns. Since the fiber length did not have a significant effect on the soundabsorption properties of the material [40], it was suitable for the preparation of sound-absorbing materials, although the length of waste hemp fibers varied widely.

**Table 3.** Length distribution of waste hemp fibers.

| Length Range/mm | Number of Roots/Root |
| --- | --- |
| 20–40 | 67 |
| 40–60 | 27 |
| 60–75 | 6 |

Fiber fineness—The microscopic image of the waste hemp fiber was obtained by using an upright metallographic microscope. The microscopic shape of the fiber was clearly displayed on the computer by adjusting the objective magnification, and then the diameter of the waste hemp fiber was obtained. The measured image of some waste hemp fibers is shown in Figure 5.

As can be seen in Figure 5, the diameter distribution of waste hemp fiber was not uniform, and the fiber was part of the round tube and the flat band, which made the fiber fineness more uneven. The fineness of the waste hemp fibers obtained after multiple measurements was 19–38 µm, with an average diameter of 27.29 µm. The average diameters of cotton fiber, wool fiber, and polyester fiber are 20.19 µm, 32.89 µm, and 36.35 µm, respectively. It could be seen that the fineness of waste hemp fiber was larger than that of cotton fiber and smaller than wool fiber and polyester fiber. As the fiber fineness decreased, the number of fiber roots increased accordingly when the bulk density of the fiber aggregates was certain, resulting in a more tortuous path and greater airflow resistance within the fiber aggregate. At the same time, the chance of contact between fine fibers and acoustic waves increased, and the frictional viscosity caused by air vibration

increased the airflow resistance of the fiber aggregate and improved the sound-absorbing performance [41]. Because fine fibers move more easily compared to coarse fibers, a fiber aggregate composed of finer fibers would convert acoustic energy into thermal energy more quickly. In addition, the presence of fine fibers could reduce the chance of pore connectivity, thus improving the acoustic performance of the fiber aggregate [42]. Therefore, the smaller fineness of the waste hemp fiber was conducive to the improvement of the sound absorption performance of the waste hemp fiber.

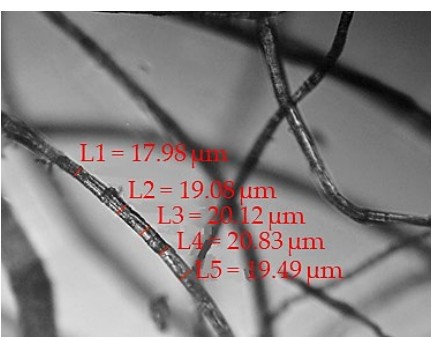 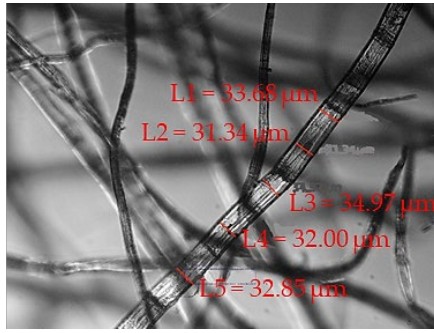

**Figure 5.** Diameter measurement graph of waste hemp fibers.

Transverse and longitudinal morphology of waste hemp fibers—Scanning electron microscope photos of the transverse and longitudinal morphological structures of the discarded hemp fibers are shown in Figures 6 and 7.

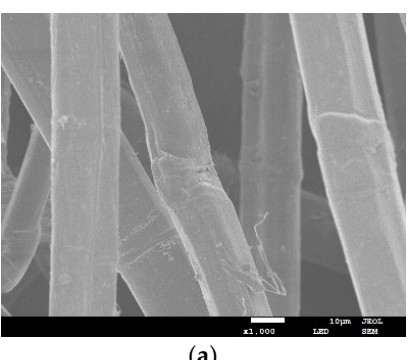 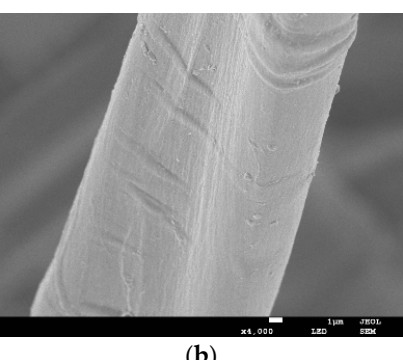 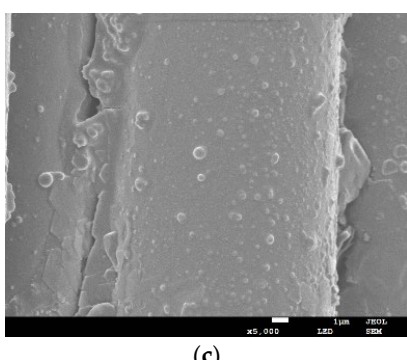

(**a**)        (**b**)        (**c**)

**Figure 6.** Scanning electron microscope photos of the longitudinal morphological structure of waste hemp fiber: (**a**) longitudinal irregular shape of the waste hemp fiber (×200); (**b**) rough structure of the surface of the waste hemp fiber (×900); (**c**) particle structure of the surface of the waste hemp fiber (×4000).

Figure 6a indicates that the longitudinal thickness of waste hemp fiber is not uniform, with one portion having a round tube shape and the other part a flat band shape. Figure 6b demonstrates that the surface of waste hemp fiber seems to be rough and irregular in shape, with the existence of surface gaps, holes, a transverse knife-cut pattern, and no natural bending. Figure 6c reveals the presence of some raised particulate and irregular cracks on the fiber surface. Therefore, the longitudinal surface of the fiber was not smooth, and when the acoustic waves were incident on the longitudinal surface of the waste hemp fiber, the presence of particles, gaps, cracks, holes, and knife-cut patterns on the fiber surface expanded the specific surface area of the fiber, thus increasing the contact area of the acoustic waves with the fiber, and the larger the specific surface area of the fiber, the better the sound-absorbing effect [43].

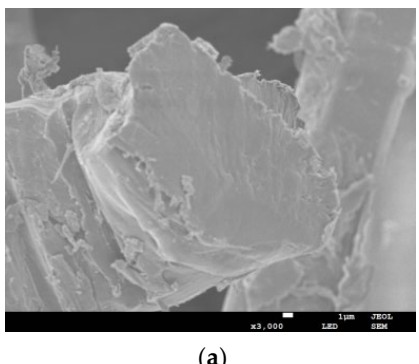
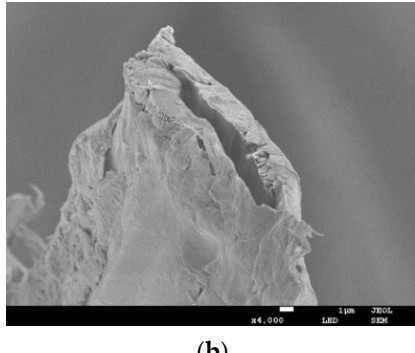
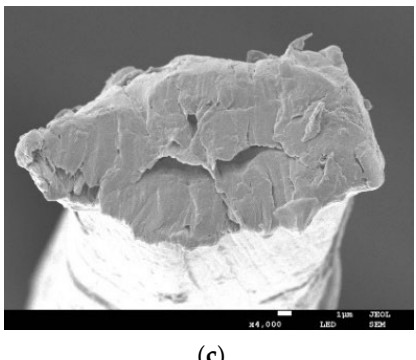

(**a**)　　　　　　　　　　　　　　(**b**)　　　　　　　　　　　　　　(**c**)

**Figure 7.** Scanning electron microscope photos of transverse morphology of waste hemp fiber: (**a**) head-end morphology of waste hemp fibers (×3000); (**b**) cavity structure of the flattened ribbon-like part of waste hemp fibers (×4000); (**c**) cavity structure of the tubular part of waste hemp fibers (×4000).

Figure 7a demonstrates that the face end of the waste hemp fiber is in a blocked state, and the cross-sectional shape is irregular. Figure 7b shows that there is a cavity in the flat ribbon part of the waste hemp fiber, and the degree of hollowness is large. Figure 7c shows that there is also a cavity in the round tubular part of the waste hemp fiber, but the hollow degree is small and the shape is irregular. Therefore, there were randomly distributed irregular cavities in the waste hemp fiber. The sound propagated in the fibers. It caused the air in the cavity inside the fiber to oscillate and caused friction between both the air and the fiber wall, converting sound energy into heat and mechanical energy and dissipating it. The shape of the cross-section influenced the sound-absorbing effectiveness, and complex and tedious shapes improved the sound absorption performance [44], while the heterogeneous cross-section of waste hemp fiber increased the contact area with sound waves, expanded the range of absorption of sound energy, and had a good dissipation effect on sound waves.

### 3.2. Test Results and Analysis of Sound-Absorbing Performance

#### 3.2.1. Test Results and Analysis of the Sound-Absorbing Performance of Fiber Aggregates

An impedance tube transfer function approach was employed to determine the absorption properties of fiber aggregates at various frequencies. The average sound-absorbing coefficient and noise reduction coefficient (*NRC*) of each fiber aggregate are reported in Table 4. To evaluate the sound-absorbing performance of fiber aggregates, the average sound-absorbing coefficient and *NRC* were utilized as measurements. From the table, it is seen that the sound-absorbing performance of the four fiber aggregates are in the following order: cotton fiber, waste hemp fiber, wool fiber, and polyester.

**Table 4.** Average sound-absorbing coefficient, noise reduction coefficient, and sound absorption coefficient of fiber aggregates at each frequency.

| Sample Code | Type of Fiber | 125 Hz | 250 Hz | 500 Hz | 1000 Hz | 2000 Hz | 4000 Hz | $\alpha$ | *NRC* |
|---|---|---|---|---|---|---|---|---|---|
| a | Cotton fiber | 0.07 | 0.09 | 0.12 | 0.40 | 0.45 | 0.86 | 0.33 | 0.25 |
| b | Waste hemp fiber | 0.07 | 0.11 | 0.15 | 0.30 | 0.37 | 0.73 | 0.29 | 0.25 |
| c | Wool fiber | 0.11 | 0.11 | 0.11 | 0.26 | 0.32 | 0.58 | 0.25 | 0.20 |
| d | Polyester | 0.09 | 0.11 | 0.12 | 0.22 | 0.3 | 0.49 | 0.22 | 0.20 |

The sound-absorbing coefficient curves of different kinds of fiber aggregates surveyed by the test are seen in Figure 8. As can be seen in Figure 8, the sound-absorbing coefficient curves of the four fiber aggregates had similar trends, i.e., the absorption coefficient increases gradually with the increase in frequency, among which the high-frequency sound-absorbing performance of the waste hemp fiber was the best, with a maximum

sound-absorbing coefficient of 0.95 and the corresponding sound-absorbing frequency of 5000 Hz.

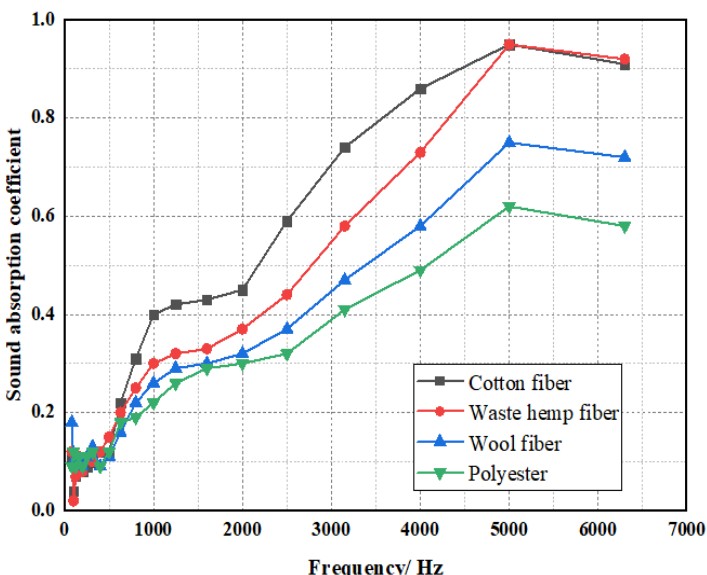

**Figure 8.** The sound-absorbing coefficient curve of the fiber aggregate.

As can be seen from Figure 8, the high-frequency sound absorption performance of waste hemp fiber was much higher than that of the low-frequency one, because the volume fraction of waste hemp fiber was small, under the same bulk density conditions. Under the condition of a certain volume density, the number of fiber roots of the discarded hemp fiber was small. When a sound wave was transmitted to the surface of a fiber aggregate, the low-frequency sound wave had a longer wavelength and was soaked up primarily by the internal material, allowing the low-frequency sound wave to effortlessly bypass the fibers, resulting in fiber absorption of less sound energy, making the low-frequency absorption coefficient lower [45,46]. Nonetheless, the high-frequency sound wave was shorter and could not bypass the fibers. When sound waves act on fiber aggregates, resonance friction will occur among both fibers and also between fibers and air. Since there was a certain cavity inside the discarded waste hemp fiber when the sound wave was introduced to the fiber interior, the air column in the cavity will vibrate, and at the same time, there will be friction amongst the air column and the fiber wall. The energy released from the air is changed to heat energy due to viscous resistance, so these vibrations and friction are components of the sound energy into kinetic energy and heat energy dissipation, resulting in the waste hemp fiber's high-frequency sound absorption performance being excellent with a high absorption coefficient [47].

Table 5 shows the relevant parameters of the tested fibers. From the table, it can be seen that the volume fraction of cotton fibers was the smallest, but the length and fineness were smaller, so in the same bulk density in the same conditions, the number of roots of cotton fibers is more, the porosity between fibers was higher, while the longitudinal existence of a curved band of cotton fibers meant that the same cotton fiber had a certain hollow structure inside. When the sound energy was acting on the fiber aggregate, the friction effect between the fibers, air vibration, and the air column inside the fiber consumed more sound energy, making the cotton fiber the best-performing sound absorption material. Similarly, the sound absorption performance of waste hemp fiber was better than that of wool fiber and polyester fiber. Wool fiber had a higher volume fraction than polyester fiber, making the number of wool fibers more than polyester fibers under the same bulk density conditions. At the same time, the wool fiber surface's unique scale-like curl structure caused the pore curvature inside the fiber aggregate to rise, the friction among fibers to be enhanced, and the contact area between fibers and air to increase, which led to more

sound energy being transformed into the other energy and consumed; therefore, the sound-absorbing performance of wool was better than polyester fiber. As a consequence, the fiber aggregate sound-absorbing performance from high to low order was cotton fiber, waste hemp fiber, wool fiber, and polyester fiber.

**Table 5.** Related parameters of measured fiber.

| Fiber Tape | Fiber Density/(g/cm$^3$) | Fiber Volume Fraction/% | Fiber Fineness/μm | Fiber Length/mm |
|---|---|---|---|---|
| Cotton fiber | 1.58 | 4.029 | 16–22 | 25–35 |
| Waste hemp fiber | 1.50 | 4.244 | 19–38 | 20–50 |
| Wool fiber | 1.32 | 4.832 | 10–50 | 30–70 |
| Polyester fiber | 1.38 | 4.613 | 28–40 | 51–76 |

### 3.2.2. Test Results and Analysis of the Sound-Absorbing Performance of Waste Hemp Fiber Composite Material

Polylactic acid (PLA) is a derivative of lactic acid (LA) produced from renewable resources such as wheat, rice straw, corn, and sorghum and has good biodegradability in the natural environment [48]. The waste hemp/PLA fiber composite made by hot pressing has the excellent properties of being green, degradable, and renewable in both components. The sound-absorbing coefficient curves of the waste hemp fiber composite material experimental results are shown in Figure 9. As shown in Figure 9, the composite material made of waste hemp fiber and PLA fiber retained the excellent characteristics of the high-frequency sound-absorbing performance of waste hemp fiber while improving the low- and medium-frequency sound-absorbing performance of the composite material. This was due to the softening and melting of PLA fiber during hot pressing that was uniformly dispersed inside the composite material, which formed a good adhesion effect between the hemp fiber and the internal structure of the material, tending to stabilize it. Therefore, the low and medium frequency sound waves did not bypass the fibers, increasing the friction between sound waves and fibers and air in the process of incidence, thus consuming energy and improving the sound-absorbing performance of the composite material. When the acoustic frequency was 5000 Hz, the absorption coefficient of waste hemp fiber composites reached a maximum of 0.93; the average absorption coefficient was 0.35; and the noise reduction coefficient was 0.30, with sound absorption properties, indicating that waste hemp fiber could be used for the preparation of acoustic composites.

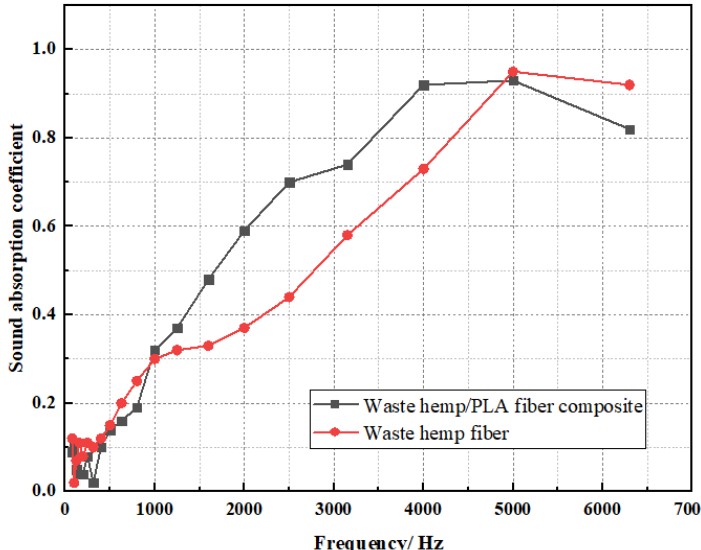

**Figure 9.** The sound-absorbing coefficient curve of waste hemp/PLA fiber composite and waste hemp fiber.

## 4. Conclusions

The research object in this paper was waste hemp fiber, and the relationship between both the microstructure of waste hemp fiber and its sound-absorbing performance was discussed, leading to the following results.

The waste hemp fiber's macromolecular structure, supramolecular structure, and morphological structure were studied. The results showed that when the sound energy hit the waste hemp fiber, due to the friction effect of the rigid oxygen hexa-ring structure of the waste hemp fiber macromolecules, the macromolecular interactions, the thermal conductivity of macromolecular chain segments, the unique hollow structure of the fiber, and the fiber's unsmooth surface would convert the sound energy into heat and mechanical energy dissipation so that the waste hemp fiber produced the sound-absorbing effect.

The impacts of various types of fiber aggregates on their sound-absorbing properties and the sound-absorbing characteristics of waste hemp fiber composites were investigated. The analysis indicated that the measured sound-absorbing characteristics of the fiber aggregates were, in descending order, cotton fiber, waste hemp fiber, wool fiber, and polyester. Additionally, the waste hemp fiber aggregate's high-frequency sound-absorbing performance was excellent, with a maximum sound-absorbing coefficient of 0.95 at a frequency of 5000 Hz. At the same time, the waste hemp fiber composite material retained the excellent high-frequency sound-absorbing properties of waste hemp fiber while improving the low- and medium-frequency sound-absorbing performance. The highest sound-absorbing coefficient of the composite material reached 0.93, which illustrates the broad prospect of waste hemp fiber for the preparation of sound-absorbing composite materials.

The four fiber aggregates studied in this paper were loosely packed fibers without compression treatment, which elucidated the fundamental reason why waste hemp fibers possess sound-absorbing properties and their sound-absorbing mechanism. This provides a new way for the efficient and high-value recycling of waste hemp fibers and provides a theoretical and experimental basis for the preparation of sound-absorbing materials with high absorption coefficients, low costs, and green degradability that are environmentally friendly and non-polluting, while contributing to the goal of peak carbon emissions and carbon neutrality. In addition, the paper only briefly introduced the waste hemp/PLA fiber composites with the sound-absorbing properties without studying the specific optimization process and sound-absorbing principle of the composites, which laid a good foundation for future research on the sound-absorbing properties of waste hemp/PLA fiber composites.

**Author Contributions:** Data curation, D.Z.; Software, X.Z. and Y.G.; Writing—original draft, D.Z.; Writing—review and editing, L.L. All authors have read and agreed to the published version of the manuscript.

**Funding:** This research was funded by the Science and Technology Innovation Fund Project of Dalian, grant number 2019J12SN71.

**Institutional Review Board Statement:** Not applicable.

**Informed Consent Statement:** Not applicable.

**Data Availability Statement:** Not applicable.

**Acknowledgments:** The authors would like to thank the Science and Technology Innovation Fund Project of Dalian (2019J12SN71) for funding parts of this work.

**Conflicts of Interest:** The authors declared no potential conflict of interest with respect to the research, authorship and/or publication of this article.

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
