# Peer review of "Structural Characteristics and Sound Absorption Properties of Waste Hemp Fiber"

_coatings, doi:10.3390/coatings12121907_

Round 1
Reviewer 1 Report
In the presented manuscript "Structural Characteristics and Sound Absorption Properties of Waste Hemp Fiber", the authors consider the use of hemp waste to obtain soundproof materials. It should be noted right away that a couple of days ago another work of the authors was published - https://www.mdpi.com/2073-4360/14/22/4844 (Structure of Waste Hemp Stalks and Their Sound Absorbing Properties), which is devoted to a similar topic. Hence, the novelty of the presented work is significantly reduced.
Authors should show the advantages of this work in comparison with the already published one.
The work was done with numerous mistakes, blots, etc.
Many drawings need to be redone and improved in quality.
There is no pagination, which complicates the work with the manuscript.
Figure 1, 5, 6. There is no need to provide these figures. I recommend to delete.
Figure 7. The figure needs to be corrected.
Fibers with a length of 20-75 (20-40) mm can be processed into textiles, why do the authors classify them as waste?
Figure 8. In many papers devoted to the study of the properties of hemp fibers, the average diameter (50-150 mkm) is significantly larger compared to those presented in this paper. How do the authors explain this?
Figure 9. The quality of the figure must be improved.
Figure 11. On the presented dependence, many values ​​are close, it is necessary to indicate the range of values ​​obtained for a particular fiber, measurement error, etc.).
Figure 12. In my opinion, it is necessary to add data for PLA to the figure.
Author Response
Authors should show the advantages of this work in comparison with the already published one.
Response: Thank you for your suggestion. The material discussed in the published papers is waste hemp straw, while this paper is a study of the structure of waste hemp fibers and their sound absorption properties. In order to provide a new research direction for the recycling of hemp fiber and provide the theoretical and experimental basis for the preparation of hemp fiber sound-absorbing composites, this paper took waste hemp fiber (long hemp spinning process generated by the fallen hemp fiber) as the research object, analyzed the relationship between the macromolecular structure of waste hemp fiber, the supramolecular structure and their morphological structure and their sound-absorbing properties. The sound-absorbing properties of fiber aggregates and their causes were studied. It was determined that the waste hemp fibers could be used for the preparation of sound-absorbing composites.
The work was done with numerous mistakes, blots, etc.
Response: We are sorry for that. We have revised the paper.
Many drawings need to be redone and improved in quality.
Response: We are sorry for this. We have made changes in the paper.
There is no pagination, which complicates the work with the manuscript.
Response: Thank you for your suggestion very much. We have revised the paper.
Figure 1, 5, 6. There is no need to provide these figures. I recommend to delete.
Response: Thank you for your suggestion. We have removed Figures 1,5,6 from the paper.
Figure 7. The figure needs to be corrected.
Response: Thank you for your suggestion. We have changed it in this paper.
Fibers with a length of 20-75 (20-40) mm can be processed into textiles, why do the authors classify them as waste?
Response: According to the suggestion, “why do the authors classify them as waste ”. Because the hemp fiber used in this paper is the long hemp spinning of fallen hemp fiber, so the short pile rate is higher, the fiber length uneven rate is higher, spinning is more difficult, so it is called waste hemp fiber.
Figure 8. In many papers devoted to the study of the properties of hemp fibers, the average diameter (50-150 mkm) is significantly larger compared to those presented in this paper. How do the authors explain this?
Response:The hemp fiber used in the paper is long hemp spinning of fallen hemp fiber, after degumming, and this treatment measures will reduce the fineness of hemp fiber, so the diameter of the waste hemp fiber in this paper is smaller.
Figure 9. The quality of the figure must be improved.
Response: Thank you for your suggestion. We have changed it in this paper.
Figure 11. On the presented dependence, many values ​​are close, it is necessary to indicate the range of values ​​obtained for a particular fiber, measurement error, etc.).
Response: Thank you for your suggestion. The data in Figure 11 have been illustrated in Tables 3.
Figure 12. In my opinion, it is necessary to add data for PLA to the figure.
Response: Thank you for your suggestion. We have changed it in this paper.
Reviewer 2 Report
Comments from Reviewer:
"According to the World Health Organization (WHO), more than 1.6 million people in Western Europe die each year from traffic noise [2]." - this citation is not accurate and does not correspond to reality.
"The strains with a THC content of less than 0.3 percent are classified as industrial hemp and can be grown and used [11]" - This statement cannot be generalized. It depends on the national legislation.
"...the largest producers of industrial hemp currently being Europe, China, and Canada [12]" - comparing Europe as a continent with the countries of China and Canada is geographically incorrect.
The explanatory notes to formulas (1) and (2) lack indexes and italics. Indexes are not in whole paper.
"...diffractive intensity when 2θ is 18°." (2θ is 20).
"The capability of sound-absorbing materials is mainly determined by the average sound absorption coefficient (α) and noise reduction coefficient (NRC). For materials, when the average sound absorption coefficient was greater than 0.2, it can be called a sound-absorbing material. When the average sound absorption coefficient was greater than 0.56, they are known as high-efficiency sound-absorbing materials [28]." - What other source (besides 28) does this claim refer to? According to my knowledge, due to the frequency dependence of the sound absorption coefficient, the best evaluation is the graphic output. It is also suitable for comparing different samples. If we only want to compare the sound absorption of different materials in a single number, NRC is used. I did not find Average absorption coefficient in technical practice, so I recommend keeping only NRC in this paper.
Table 3 – NRC is rounded to the nearest 0.05…
“…the noise reduction coefficient was 0.245, with sound absorption properties, indicating that waste hemp fiber could be used for the preparation of acoustic composites” - NRC must be rounded to 0.25. Do you think that the value of 0.25 is sufficient for the preparation of acoustic composites? This value places the material in the lowest acoustic absorption class.
Figure 12 could contain the graph of waste hemp fiber from Figure 11.
I cannot agree with the conclusions of the article ("This paper clarified the fundamental reason for the good sound absorption performance of waste hemp fiber and its sound-absorbing mechanism."). Based on the values mentioned, the sound-insulating properties of waste hemp fibers are relatively weak in normally audible frequencies. At higher frequencies, most materials are good enough.
Author Response
"According to the World Health Organization (WHO), more than 1.6 million people in Western Europe die each year from traffic noise [2]." - this citation is not accurate and does not correspond to reality.
Response: We are sorry for this mistake. We have removed the reference in the paper.
"The strains with a THC content of less than 0.3 percent are classified as industrial hemp and can be grown and used [11]" - This statement cannot be generalized. It depends on the national legislation.
Response: According to the suggestion, “This statement cannot be generalized. It depends on the national legislation”. We have added it in paper.The THC level of various cannabis strains varies significantly, but according to the 2018 U.S Farm Bill, the strains with a THC content of less than 0.3 percent are classified as industrial hemp and can be grown and used.
"...the largest producers of industrial hemp currently being Europe, China, and Canada [12]" - comparing Europe as a continent with the countries of China and Canada is geographically incorrect.
Response: We are sorry for this mistake. We have changed it in the paper. At least 47 nations throughout the world presently cultivate cannabis for business or research purposes, with the largest producers of industrial hemp currently being China, Canada, and France.
The explanatory notes to formulas (1) and (2) lack indexes and italics. Indexes are not in whole paper.
Response: We are sorry for this mistake. We have made changes in the paper. XC is crystallinity, I002 is the maximum intensity of the lattice diffractive peak, Iam is the diffractive intensity of the amorphous phase, and the background diffractive intensity when 2θ is 20°. Vf is the fiber volume fraction (%); m is the mass of fiber aggregate (g); ρ is the fiber density (g/cm3); V0 is the volume of fiber aggregate (cm3).
"...diffractive intensity when 2θ is 18°." (2θ is 20).
Response: According to the suggestion, “"...diffractive intensity when 2θ is 18°." (2θ is 20)”. We have changed it in this paper.
"The capability of sound-absorbing materials is mainly determined by the average sound absorption coefficient (α) and noise reduction coefficient (NRC). For materials, when the average sound absorption coefficient was greater than 0.2, it can be called a sound-absorbing material. When the average sound absorption coefficient was greater than 0.56, they are known as high-efficiency sound-absorbing materials [28]." - What other source (besides 28) does this claim refer to? According to my knowledge, due to the frequency dependence of the sound absorption coefficient, the best evaluation is the graphic output. It is also suitable for comparing different samples. If we only want to compare the sound absorption of different materials in a single number, NRC is used. I did not find Average absorption coefficient in technical practice, so I recommend keeping only NRC in this paper.
Response: Thank you for your suggestion, this claim also comes from “Structural characteristics and sound absorption properties of poplar seed fibers.( DOI: 10.1177/0040517520921396)” and “Structure of Waste Hemp Stalks and Their Sound Absorbing Properties (https://www.mdpi.com/2073-4360/14/22/4844)”. In this paper, the sound absorption performance of waste hemp fiber is compared with other three fibers, and the average absorption coefficient and noise reduction coefficient are chosen as the basis for comparison in order to better illustrate the sound absorption performance of waste hemp fiber.
Table 3 – NRC is rounded to the nearest 0.05…
Response: We are sorry for this mistake. We have changed it in the paper.
“…the noise reduction coefficient was 0.245, with sound absorption properties, indicating that waste hemp fiber could be used for the preparation of acoustic composites” - NRC must be rounded to 0.25. Do you think that the value of 0.25 is sufficient for the preparation of acoustic composites? This value places the material in the lowest acoustic absorption class.
Response: According to the suggestion, “This value places the material in the lowest acoustic absorption class”. The sound absorption performance of this composite material has been re-experimented, and its average absorption coefficient was measured to be 0.35 and NRC to be 0.30 to meet the minimum requirement of sound absorption material, and the subsequent research can optimize its hot pressing and preparation process so as to further improve the sound absorption performance of the composite material, so this material can be used for the preparation of sound absorption composite material.
Figure 12 could contain the graph of waste hemp fiber from Figure 11.
Response: According to the suggestion, “Figure 12 could contain the graph of waste hemp fiber from Figure 11.”. We have changed it in this paper.
I cannot agree with the conclusions of the article ("This paper clarified the fundamental reason for the good sound absorption performance of waste hemp fiber and its sound-absorbing mechanism."). Based on the values mentioned, the sound-insulating properties of waste hemp fibers are relatively weak in normally audible frequencies. At higher frequencies, most materials are good enough.
Response: We are sorry for this mistake. We have changed it in the paper. This paper clarified the fundamental reason for the sound absorption performance of waste hemp fiber and its sound-absorbing mechanism. According to the experiment in this paper, the average sound absorption coefficient and noise reduction coefficient of waste hemp fiber are close to those of cotton fiber, while cotton fiber has been widely used in the field of sound absorption, and its maximum sound absorption coefficient can reach 0.95, so it has sound absorption performance and can be used in the field of sound absorption.
Reviewer 3 Report
Manuscript ID: coatings-2045212
Title: Structural characteristics and sound absorption properties of waste hemp fiber
Authors: Zhang et al.
In this manuscript, authors have characterized the waste hemp fiber and its application in sound absorption. Authors claimed that the waste hemp fiber had excellent high-frequency sound-absorbing performance, with a maximum sound absorption coefficient of 0.95, compared with other type of fibers such as cotton fiber, waste hemp fiber, wool fiber, and polyester fiber.
In my opinion, the novelty of this manuscript is ok, and I am willing to accept this appear after answering major revision comments as follow:
1. Kindly, modify and correct following figures:
Figure 7. X-ray diffraction pattern of waste hemp fiber. The x and y axis title are duplicated!
Figure 4. Infrared spectroscopy of waste hemp fiber. The x and y axis title are duplicated!
2- The finding of this manuscript should be compared with other reported similar studies. I suggest authors to create a table summary for comparing results of this manuscript with other reported results published elsewhere.
3- please correct and modify Table 4. Related parameters of measured fiber. Waste hemp fiber is written two times. The Wool fiber is missing!
4- kindly, correct and modify English through the whole manuscript
Author Response
In this manuscript, authors have characterized the waste hemp fiber and its application in sound absorption. Authors claimed that the waste hemp fiber had excellent high-frequency sound-absorbing performance, with a maximum sound absorption coefficient of 0.95, compared with other type of fibers such as cotton fiber, waste hemp fiber, wool fiber, and polyester fiber.
Response: Yes. You are right. Thank you for your understanding.
In my opinion, the novelty of this manuscript is ok, and I am willing to accept this appear after answering major revision comments as follow:
- Kindly, modify and correct following figures:
Figure 7. X-ray diffraction pattern of waste hemp fiber. The x and y axis title are duplicated!
Figure 4. Infrared spectroscopy of waste hemp fiber. The x and y axis title are duplicated!
Response: We are sorry for that. We have modified Figure 4. and Figure 7. in the paper.
2- The finding of this manuscript should be compared with other reported similar studies. I suggest authors to create a table summary for comparing results of this manuscript with other reported results published elsewhere.
Response: According to the suggestion, “I suggest authors to create a table summary for comparing results of this manuscript with other reported results published elsewhere”. We have created Table 1 A summary of studies of hemp fiber sound absorbers. in the paper.
3- please correct and modify Table 4. Related parameters of measured fiber. Waste hemp fiber is written two times. The Wool fiber is missing!
Response: We are sorry for that. We have revised the Table 4. in the paper.
4- kindly, correct and modify English through the whole manuscript
Response: We are very sorry for that. We have revised the paper.
Round 2
Reviewer 1 Report
The authors answered the questions posed and took into account previous comments.
However, the scientific novelty of the presented work leaves me in doubt.
The quality of most of the figures remains at a low level and needs to be corrected.
I have doubts and questions about how the authors got Fiber density/(g/cm3) 1.5-1.58 for hemp and cotton.
The technique described in the paper is unlikely to provide such values.
Should I use another method here, like a gradient column (a description of the methodology can be found, for example, here - https://doi.org/10.1007/s10692-017-9786-x)?!
In my opinion, the authors should pay more attention to the presented manuscript before its publication!
If the editor believes that the corrections made are sufficient and the scientific novelty meets the requirements of the journal, then the work can be considered for publication.
Author Response
The authors answered the questions posed and took into account previous comments.
However, the scientific novelty of the presented work leaves me in doubt.
Response: The main purpose of this paper is to explore the relationship between the structural characteristics of waste hemp fiber and its sound absorption performance and to verify that the waste hemp/polylactic acid fiber composite made of waste hemp fiber has sound absorption performance, so as to expand the application field of hemp fiber and improve the recycling of waste hemp fiber.
The quality of most of the figures remains at a low level and needs to be corrected.
Response: We are sorry for that. We have revised the paper
I have doubts and questions about how the authors got Fiber density/(g/cm3) 1.5-1.58 for hemp and cotton.
The technique described in the paper is unlikely to provide such values.
Should I use another method here, like a gradient column (a description of the methodology can be found, for example, here - https://doi.org/10.1007/s10692-017-9786-x)?!
Response: We have added the testing method of fiber density in the page 6 of the article. Density of fiber aggregates. The density of fiber aggregates was determined using a gradient column. The method was based on comparing the immersion depth of the fiber spheres under test relative to the density standard of glass spheres in a cylinder with liquid, and the density of the fiber could be found by checking the density height graph according to the height of the fiber spheres.
In my opinion, the authors should pay more attention to the presented manuscript before its publication!
Response: Thank you for this suggestion very much. We have revised the paper.
If the editor believes that the corrections made are sufficient and the scientific novelty meets the requirements of the journal, then the work can be considered for publication.
Response: Thank you for all your suggestions.
Reviewer 2 Report
The paper was improved, but I have some comments.
Page 5: "the background diffractive intensity when 2θ is 20°" - I do not understand this explanation when 2θ is 20°" I don't understand this explanation. The character θ must be replaced by 0 in the text.
Page 13: "the average absorption coefficient was 0.35, and the noise reduction coefficient was 0.30, with sound absorption properties, indicating that waste hemp fiber could be used for the preparation of acoustic composites." These modified values do not correspond to the results in Table 4. The values in this table are calculated correctly, so I would also use them in the text.
Page 13: "The sound-absorbing coefficient curves of the waste hemp fiber composite material experimental results are shown in Figure 12." - Figure 12 this version of the paper no longer contains (similarly figure 11 on page 12) - the whole article should be checked.
Page 14 : "Besides that, the waste hemp fiber aggregate's high-frequency sound-absorbingperformance was excellent, with a maximum sound-absorbing coefficient of 0.95". - if you want to use this sentence in the conclusion, it is necessary to add the frequency at which this value was reached (similarly at The highest sound-absorbing coefficient of the composite material reached 0.93).
The last paragraph in the conclusion is incomprehensible - please rephrase it.
Author Response
Page 5: "the background diffractive intensity when 2θ is 20°" - I do not understand this explanation when 2θ is 20°" I don't understand this explanation. The character θ must be replaced by 0 in the text.
Response: According to the suggestion, “The character θ must be replaced by 0 in the text.”. We have changed it in this paper. Iam is the diffractive intensity of the amorphous phase, and the background diffractive intensity is 20°.
Page 13: "the average absorption coefficient was 0.35, and the noise reduction coefficient was 0.30, with sound absorption properties, indicating that waste hemp fiber could be used for the preparation of acoustic composites." These modified values do not correspond to the results in Table 4. The values in this table are calculated correctly, so I would also use them in the text.
Response: Thank you for this suggestion. The average absorption coefficient and noise reduction coefficient stated in the paragraph on page 13 are both the average absorption coefficient and noise reduction coefficient of the waste hemp/PLA composite calculated from Figure 9, while the data in Table 4 are the average absorption coefficient and noise reduction coefficient of the waste hemp fiber calculated from Figure 8, so these two data are not consistent.
Page 13: "The sound-absorbing coefficient curves of the waste hemp fiber composite material experimental results are shown in Figure 12." - Figure 12 this version of the paper no longer contains (similarly figure 11 on page 12) - the whole article should be checked.
Response: We are sorry for this mistake. We have changed Figure 12 to Figure 9. And we have checked whole article.
Page 14 : "Besides that, the waste hemp fiber aggregate's high-frequency sound-absorbingperformance was excellent, with a maximum sound-absorbing coefficient of 0.95". - if you want to use this sentence in the conclusion, it is necessary to add the frequency at which this value was reached (similarly at The highest sound-absorbing coefficient of the composite material reached 0.93).
Response: Thank you for this suggestion, we have added it in the conclusion. Besides that, the waste hemp fiber aggregate's high-frequency sound-absorbing performance was excellent, with a maximum sound-absorbing coefficient of 0.95 at a frequency of 5000 Hz.
The last paragraph in the conclusion is incomprehensible - please rephrase it.
Response: We are sorry for that. We have revised it. The four fiber aggregates studied in this paper were loosely packed fibers without compression treatment, which elucidated the fundamental reason why waste hemp fibers possess sound-absorbing properties and their sound-absorbing mechanism. This provided a new way for the efficient and high-value recycling of waste hemp fibers and provided a theoretical and experimental basis for the preparation of sound-absorbing materials with high absorption coefficients, low costs, and green degradability that are environmentally friendly and non-polluting, while contributing to the goal of peak carbon emissions and carbon neutrality. In addition, the paper only briefly introduced the waste hemp/PLA fiber composites with sound-absorbing properties without studying the specific optimization process and sound-absorbing principle of the composites, which laid a good foundation for future research on the sound-absorbing properties of waste hemp/PLA fiber composites.

Reviewer 3 Report
The revised manuscript has been improved. Authors have answered all review comments. Thus, I recommend to accept this manuscript in present form
Author Response
The revised manuscript has been improved. Authors have answered all review comments. Thus, I recommend to accept this manuscript in present form
Response: Thank you for your suggestion.
Round 3
Reviewer 1 Report
The authors of the manuscript have not yet introduced line numbering, which makes it difficult to work with it. How do the authors relate high crystallinity to the density obtained for the samples - "The crystallinity of waste hemp fiber was calculated according to Equation (1) as 48.84%, whereas the crystallinity of cotton, wool fiber, and flax fiber was 78.8% [35], 81.2 % [36], and 69.3% [37], respectively. As a result, discarded waste hemp fibers had a lower crystallinity than other natural fibers." and Table 5. "Wool fiber density/(g/cm3) - 1.32."?